# Exploring the joint probability of precipitation and soil moisture over Europe using copulas

Carmelo Cammalleri[1], Carlo De Michele[1], Andrea Toreti[2]

[1]Dipartimento di Ingegneria Civile e Ambientale (DICA), Politecnico di Milano, Milan, 20133, Italy.
[2]European Commission, Joint Research Centre (JRC), Ispra, 21027, Italy.

*Correspondence to*: Carmelo Cammalleri (carmelo.cammalleri@polimi.it)

**Abstract.** The joint probability of precipitation and soil moisture is here investigated over Europe with the goal to extrapolate meaningful insights on the potential joint use of these variables for the detection of agricultural droughts within a multivariate probabilistic modeling framework. The use of copulas is explored, being the framework often used in hydrological studies for the analysis of bivariate distributions. The analysis is performed for the period 1996-2020 on the empirical frequencies derived from ERA5 precipitation and LISFLOOD soil moisture datasets, both available as part of the Copernicus European Drought Observatory. The results show an overall good correlation between the two standardized series (Kendall's $\tau$ = 0.42±0.1), but also clear spatial patterns in the tail-dependence derived with both non-parametric and parametric approaches. About half of the domain shows symmetric tail-dependence, well reproduced by the Student-t copula; whereas the rest of the domain is almost equally split between low and high tail-dependences (both modeled with the Gumbel family of copulas). These spatial patterns are reasonably reproduced by a random forest classifier, suggesting that this outcome is not driven by chance. This study stresses how a joint use of standardized precipitation and soil moisture for agriculture drought characterization may be beneficial in areas with strong low tail-dependence, and how this behavior should be carefully considered in multivariate drought studies.

## 1. Introduction

Agricultural drought, defined as a condition of unusually high precipitation shortages and/or soil water deficits causing adverse effects on crop yields and production (Panu and Sharma, 2002), is probably the most recognized of the four main drought types or phases (Wilhite and Glantz, 1985). This is mainly due to the more direct and easier to understand impacts compared to the other types of droughts (Mishra and Singh, 2010). The scientific literature on agricultural drought provides a large variety of indices (WMO and GWP, 2016), with the aim of reproducing the temporal dynamics of crop water deficit through a combination of climatic observations, hydrological modeling, and remote sensing data (Zargar et al., 2011).

The difficulty in capturing the multi-facet nature of agricultural drought events across the world with a single approach (Sivakumar et al., 2011) is confirmed by the absence of consensus in the scientific literature on the most reliable agricultural drought index. Despite the large range of available indices, some common characteristics can be identified, such as the focus on some proxy variables of plant water availability – through soil moisture (Dutra et al., 2008), actual evapotranspiration (Anderson et al., 2011) or basic meteorological information (Vicente-Serrano et al., 2010) – and the need to account for deviations from long-term conditions (i.e., use of standardized anomalies).

Meteorological drought indicators computed on appropriate aggregation time scales (McKee et al., 1993; Vicente-Serrano et al., 2010) have demonstrated a good capability to represent agricultural drought conditions in several case studies (e.g., Bachmair et al., 2018; Mohammed et al., 2022; Tian et al., 2018). They have been successfully integrated in a number of operational drought monitoring systems, thanks to their minimal input data requirement and ease of use. Among those indices, the Standardized Precipitation Index (SPI, McKee et al., 1993) computed on short-to-medium aggregation periods (i.e., SPI-3 and -6) is often adopted as a suitable proxy variable for agricultural droughts (WMO, 2012).

As highlighted by Sheffield and Wood (2007), simplified indices for drought monitoring, such as the Palmer Drought Severity index (PDSI; Palmer, 1965) or the previously mentioned meteorological indicators, have been slowly integrated with indices directly based on modeled soil moisture data. This transition is fostered by the increasing availability worldwide of process-based hydrological models. Soil moisture percentile, or similarly standardized quantities, are often used for this scope (Mo and Lettenmeier, 2013; Xia et al., 2014). The ever-growing records of remote

sensing-based estimates of soil moisture are becoming an additional data source to support the
development of dedicated soil moisture-based drought indices (Cammalleri et al., 2017; Carrão et
al., 2016).

In the context of agricultural drought, an overall good agreement between SPI and soil
moisture indices has been demonstrated over a large range of agricultural practices, crop types and
climatic conditions. Halwatura et al. (2017) showed how SPI-3 represents a good approximation
of modeled soil moisture over three different climatic regions in eastern Australia. Sims et al.
(2002) found high correlation between short-term precipitation deficit and soil moisture variations
in North Carolina, while Ji and Peters (2003) highlighted the high correlation between SPI-3 and
vegetation growth over croplands and grasslands in the U.S. Great Plains. Wang et al. (2015)
observed a good matching between soil moisture dynamics and SPI at the scale of 1-3 months
when testing various indices over China. In Europe, Manning et al. (2018) highlighted how
precipitation is the main driver of soil moisture droughts for a set of both dry and wet sites.

In spite of the above-mentioned consistencies, the outcome of any drought analysis is
inevitably affected by the index selected to characterize drought conditions over a certain study
region, as also highlighted by Quiring and Papakryiakou (2003) in testing different indices over
the Canadian prairies. These authors suggest that a variety of drought indices should always be
tested to determine the most appropriate one for a given application. It follows that the synergy
between multiple indices can be exploited by the use of multivariate indicators (Hao and Singh,
2015), a family of approaches that encompasses a variety of merging strategies, including
combined cascading indices (Cammalleri et al., 2021a; Rembold et al., 2019), composite and
integrated approaches (Brown et al., 2008; Svoboda et al., 2002), and joint probability functions
(Bateni et al., 2018; Hao and AghaKouchak, 2013; Kanthavel et al., 2022).

The latter category, in particular, aims at capturing the complex statistical dependence
among different drought-related variables (Hao and Singh, 2015), and it has seen a growing
relevance in many hydrological applications thanks to the introduction of copula functions and
their ability to model a wide range of dependence structures (Nelsen, 2006; Salvadori et al., 2007;
Joe, 2015). In the field of drought indices, the approach proposed by Kao and Govindaraju (2010)
for the computation of the Joint Deficit Index (JDI) has been applied to a variety of drought-related
quantities over different regions, often including precipitation and soil moisture (i.e., Dash et al.,
2019; Kwon et al., 2019).

A key feature in using joint probability is the possibility to characterize the so-called tail-
dependence (TD), namely the asymptotical dependence of the extremes (Frahm et al., 2005). While
TD has received large attention in the scientific literature of hydrological extremes (e.g.,
Aghakouchak et al., 2010; Poulin et al., 2007; Serinaldi, 2008), its use is largely unexploited in
studies focusing on combined drought indices.

Studies on the marginal distribution of either precipitation or soil moisture usually adopt
the Gamma distribution for precipitation and the Beta distribution for soil moisture. The use of the
Gamma family for the implementation of the SPI at different accumulation periods has become a
standard practice in many applications (e.g., Mo and Lyon, 2015; Yuan and Wood, 2013). While
other distributions have also proven to be reliable, such as the exponentiated Weibull (Pieper et
al., 2020) and the Person Type III (Ribeiro and Pires, 2016), fitting the Gamma is still the most
adopted approach. Over Europe, Stagge et al. (2015) demonstrated how the Gamma outperformed
the other tested distributions across all accumulation periods and regions.

A more limited number of applications based on soil moisture data are available in the
scientific literature compared to SPI. The use of the Beta distribution for soil moisture data has
been introduced as early as the late '70s, with the pioneer study of Ravelo and Decker (1979),
following the consideration that soil moisture is a double-bounded quantity, ranging between
residual and saturation. Sheffield et al. (2004) successfully applied this standardization for drought
analyses over the US, while the same distribution has been adopted by Cammalleri et al. (2016)
on modeled data over Europe. Most recently, the Beta distribution was also used to characterize
the frequency of global satellite soil moisture data (Sadri et al., 2020).

Conversely, no standard approaches have been identified for the application of copulas to
model the bivariate joint distribution of precipitation and soil moisture, mainly due to the large
variety of probabilistic structures than may be observed between these two quantities. Common
fitting strategies rely on the application of various copula families to identify the optimal for each
specific site (e.g., Hao and AghaKouchak, 2013), or are based on an a-priori selection of a copula
family following empirical evidence (e.g., Dixit and Jayakumar, 2021). Independently from the
selection strategy, the adopted copula implicitly assumes an underling TD behavior, which
influence on extreme detection should be properly accounted.

A comprehensive study on the joint probabilistic dynamics of precipitation and soil
moisture is currently lacking in the scientific literature of multivariate drought modeling. Hence,
the main goal of this study is to fill this gap, by investigating the mutual relationship between the
empirical frequencies of precipitation (cumulated over 3 months, as for SPI-3) and soil moisture
datasets as available over Europe as part of the European Drought Observatory of the Copernicus
Emergency Management Service (EDO, https://edo.jrc.ec.europa.eu).
A large set of copulas is tested for this purpose across the entire European domain, to
identify an optimal modeling of the dependence especially in proximity of the tails (given its major
role in extreme detection). The spatial distribution of the results is analyzed to infer evidence of
common patterns and behavior, which may support future operational applications based on
similar parametric approaches.

**2. Materials and Methods**
**2.1 Precipitation and soil moisture datasets**
The study focuses on Europe and makes use of the dataset of indicators available over the region
as part of EDO. Precipitation data accumulated over consecutive 3-month periods are used here,
as the quantity at the base of the SPI-3 index. Hourly total precipitation maps from the ECMWF
ERA5 global atmospheric reanalysis model (https://www.ecmwf.int/en/forecasts/dataset/ecmwf-
reanalysis-v5) are collected through the Copernicus Climate Change Service (C3S,
https://climate.copernicus.eu/) and cumulated at monthly updates (no missing values are present
in the reanalysis dataset). This dataset has proven to be quite reliable over Europe for drought
analyses (e.g., Cammalleri et al., 2021b; van der Wiel et al., 2022), as it is currently employed in
near-real time as part of the operational tools of EDO. Empirical frequencies of 3-month
precipitation are derived from the rainfall records, in order to obtain a non-parametric calculation
of the standardized anomaly, SPI-3, without the possible artifact introduced by the fitting of a
theoretical distribution (i.e., Gamma distribution) (see Soľáková et al., 2014). From here on, we
will refer to this dataset as standardized precipitation.
Soil moisture records over the entire European domain are derived from the simulations of
the LISFLOOD distributed hydrological rainfall–runoff model (de Roo et al., 2000). LISFLOOD
runs in near-real time as part of the European Flood Awareness System (Thielen et al., 2009), and
it provides daily soil moisture maps for the root zone at a spatial resolution of 5-km. Daily modeled
data are averaged at monthly scale and converted into a Soil Moisture Index (SMI) as in
Seneviratne et al. (2010). The model is calibrated and validated over an extensive network of river
discharge stations following the procedure described in Arnal et al. (2019), and it has been
successfully tested for drought analyses over Europe as part of EDO for the computation of the
Soil Moisture Anomaly (SMA) index (Cammalleri et al., 2015). Similar to precipitation, empirical
frequencies are computed from the monthly soil moisture data in order to obtain a non-parametric
calculation of the standardized anomaly, SMA, thus independent from a theoretical fitting (i.e.,
Beta distribution). We will refer to this dataset as standardized soil moisture from hereafter.
In this study, data collected for the most recent 25 years (1996-2020) are used as a common
period. This period is chosen to minimize the effects of non-stationarity in precipitation records
and to avoid the inclusion of early LISFLOOD records that are affected by a lower number of
ground meteorological stations in the forcing (Thieming et al., 2022). The time series of both
standardized precipitation and soil moisture at grid-cell scale are preliminarily tested for auto-
correlation using the partial auto-correlation function (PACF, Box and Jenkins, 1976). This
analysis returned positive and statistically significant (95% confidence interval) values only at lag
= 1, suggesting a substantial absence of auto-correlation beyond what is expected for time series
with smooth temporal dynamics such as 3-month cumulative precipitation and soil moisture.
The 300 maps (12 months × 25 years) for the two standardized datasets are then spatially
interpolated on a common Lambert azimuthal equal-area (LAEA) projection on a regular grid of
5-km using the nearest neighbor algorithm. This is done to preserve the high-resolution
information of the soil moisture and by considering the smooth spatial dynamics of precipitation
accumulated over 3 months.
**2.2 Copula families**
The introduction of copulas in multivariate probability modeling has provided to hydrologists a
flexible tool to reproduce the joint probability of multiple dependent variables characterized by a
variety of marginal distributions (De Michele and Salvadori, 2003; Salvadori and De Michele,

2004).

Limiting the focus on bivariate variables, the joint probability distribution, $F$, of two
random variables ($X_1$ and $X_2$) can be expressed, thanks to the Sklar's theorem, as:
$$F(x_1, x_2) = C(F_1(x_1), F_2(x_2)) \tag{1}$$
where $F_1$ and $F_2$ are the marginal distribution of $X_1$ and $X_2$, respectively, and $C$ is the copula
function (Salvadori et al., 2007).

A large variety of parametric formulations has been introduced in the literature to explicitly
link the marginal distributions to the joint probability, with some of the most common copula
families used in hydrology belonging to the Elliptical and Archimedean copulas (Chen and Guo,
2019). Two measures of dependence play a major role in parametric copula inference. The Kendall
rank correlation coefficient ($\tau$) is commonly used as a non-parametric measure of overall ordinal
association, while the so-called Tail-Dependence (TD) coefficients (Salvadori et al., 2007) are
used to estimate the asymptotical degree of dependence in the upper and lower extremes (upper
tail-dependence, $\lambda_U$, and lower tail-dependence, $\lambda_L$, respectively). The estimation of TD non-
parametrically is not an easy task, as highlighted by Serinaldi et al. (2015), as it aims at assessing
an asymptotic behavior from a finite sample. Several formulations are proposed in the scientific
literature (see Frahm et al., 2005), and the method proposed by Schmidt and Stadtmueller (2006)
is here used to obtain non-parametric estimates of both TD coefficients.

In this study, the parametric bivariate probability of standardized precipitation and soil
moisture is assessed by using the R package "VineCopula" (Aas et al., 2009; Dissman et al., 2013).
The Akaike Information Criterion (AIC, Stoica and Selen, 2004) is used to select, for each spatial
grid cell, the best fitting copula among the wide range of families available in the package. The
main properties of some relevant copulas are reported in Table 1, as they will be useful to interpret
the successive results.

In particular, from the data in Table 1 it is important to highlight how the BB7 copula is a
combination of Joe and Clayton copulas, of which it inherits the tail-dependences, and how the
TD behavior of a copula can be inverted (i.e., the upper tail-dependence can become the lower and
*vice versa*) by simply considering the reciprocal marginals (commonly known as rotated forms,
identified by the suffix 180). Information from both non-parametric and parametric approaches are
here jointly used to discriminate between different TD behaviors.


**Table 1.** Main copulas analyzed in this study and their upper and lower tail-dependence
coefficients ($\lambda_L$ and $\lambda_U$, respectively).

| Copula | $\lambda_L$ | $\lambda_U$ |
|---|---|---|
| Gaussian | 0 | 0 |

| | | |
|---|---|---|
| Student-t | $2t_{v+1}\left(-\sqrt{v+1}\sqrt{\dfrac{1-\rho}{1+\rho}}\right)$ | $2t_{v+1}\left(-\sqrt{v+1}\sqrt{\dfrac{1-\rho}{1+\rho}}\right)$ |
| Gumbel | $0$ | $2-2^{\frac{1}{\theta}}$ |
| Clayton | $2^{\frac{-1}{\theta}}$ | $0$ |
| Joe | $0$ | $2-2^{\frac{1}{\delta}}$ |
| BB7 | $2^{\frac{-1}{\theta}}$ | $2-2^{\frac{1}{\delta}}$ |

Even if a copula is selected as the optimal based on the AIC, this does not necessarily exclude that other copulas may perform similarly. For this reason, we introduced a further test based on the relative likelihood criterion (Burnham and Anderson, 2002), $\exp\left(\dfrac{AIC_{min}-AIC_i}{2}\right)$, to establish the likelihood that an AIC value of a given copula ($AIC_i$) is significantly different than the minimum value ($AIC_{min}$) obtained for the optimal solution.

**2.3 Random forest classification of selected copulas**

The interpretation of the selected copula functions may help highlighting the transferability of the observed results over different contexts. For this reason, the observed spatial distribution of the selected copulas is analyzed through a random forest classifier (Breiman, 2001), in order to find evidence of reproducible patterns beyond simple chance.

As input features we consider a set of commonly available variables, such as: ground elevation, annual average temperature, annual total precipitation, precipitation seasonality (ratio between total precipitation in warm and cold months), annual average Normalized Difference Vegetation Index (NDVI), annual average soil moisture, and soil type. As hyperparameters for the random forest, we tuned the number of trees (ntree) and the number of features randomly sampled at each split (mtry) using the "randomForest" R package (Breiman, 2001).

**3. Results**

A preliminary analysis of the degree of correlation between the monthly standardized 3-month precipitation and soil moisture (analogous to non-parametric SPI-3 and SMA) is tested on the full

timeseries of each grid cell using the Kendall's τ, as depicted in Fig. 1 for the entire European
domain.

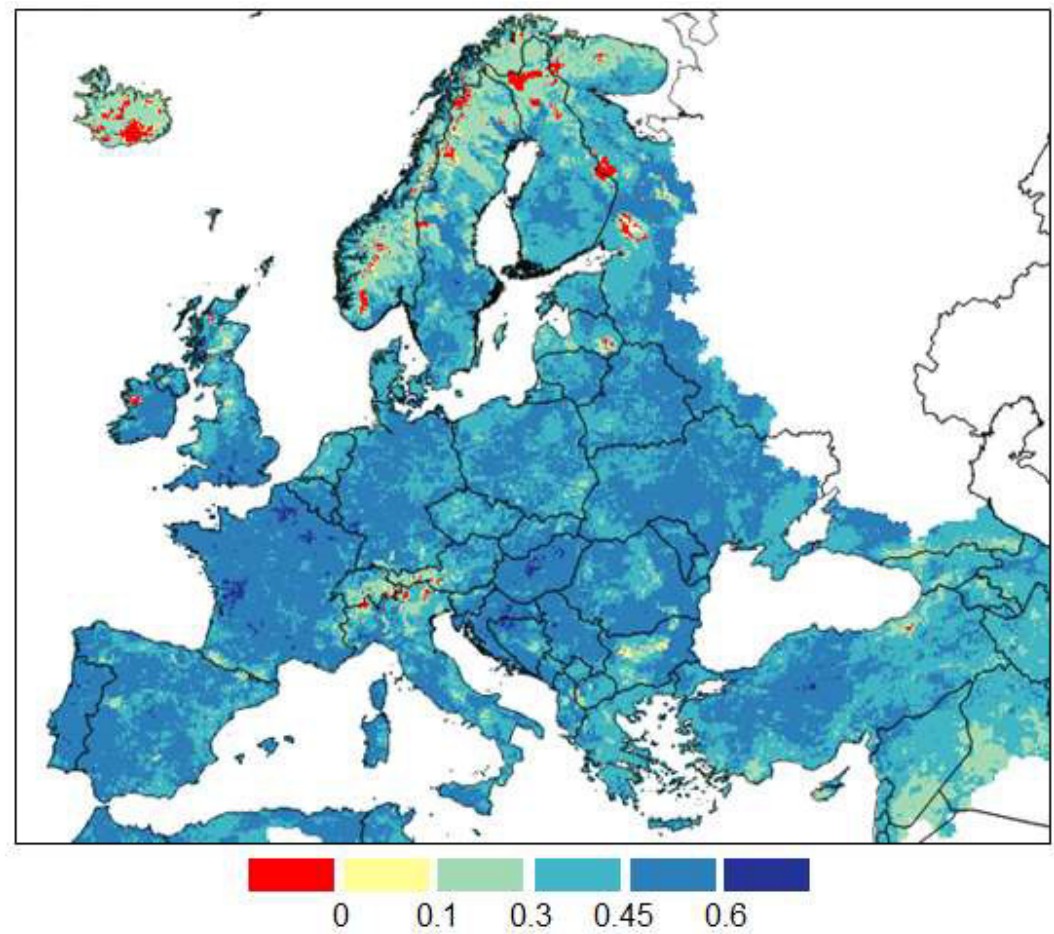


**Fig. 1.** Spatial distribution of the Kendall's τ between monthly standardized 3-month precipitation
and soil moisture. Roughly, values lower than 0.1 are not statistically significant at $p = 0.05$ (two-
tails).

The results reported in Fig. 1 confirms the expected direct relation between the two
variables, with a relatively homogeneous distribution of medium/high (between 0.3 and 0.5) τ
values (τ = 0.42±0.1). Limited regions with low (and sometimes even slightly negative) τ values
are sporadically observed, mostly over the Alps, Iceland and the coldest regions of the Scandinavia
peninsula. Low correlations over these regions are likely related to the presence of snow coverage
during extended periods of the year. Overall, the observed τ values cannot be considered
statistically significant (at $p = 0.05$) only for less than 2% of the domain.
The analysis of the non-parametric tail-dependence values is summarized in the plot
depicted in Fig. 2, where the cumulative frequency of the difference between the empirical $\lambda_L$ and
$\lambda_U$ values is reported. The range of TD values in Fig. 2 for which it is possible to exclude significant
asymmetry in the tail dependence coefficients is identified by setting a maximum value for $|\lambda_L -$
$\lambda_U|$. To define this threshold, the non-parametric TD coefficients were re-computed on shuffled
time series (to artificially reconstruct conditions of null dependence), and the $|\lambda_L - \lambda_U|$ value
corresponding to a cumulative frequency of 90% of the grid cells after the shuffling was detected
as threshold, corresponding to a value of 0.1. This value can be seen as a lower limit to identify
symmetric dependence.
The plot in Fig. 2 highlights how the majority (about 50%) of the grid cells can be
considered characterized by a symmetric behavior in the tail-dependence coefficients according to
the above mentioned criterion ($|\lambda_L - \lambda_U| < 0.1$), whereas the rest of the grid cells are almost equally
split between a predominance of the Upper Tail-Dependence (UTD, corresponding to negative
differences) or a predominance of Lower Tail-Dependence (LTD, positive differences).

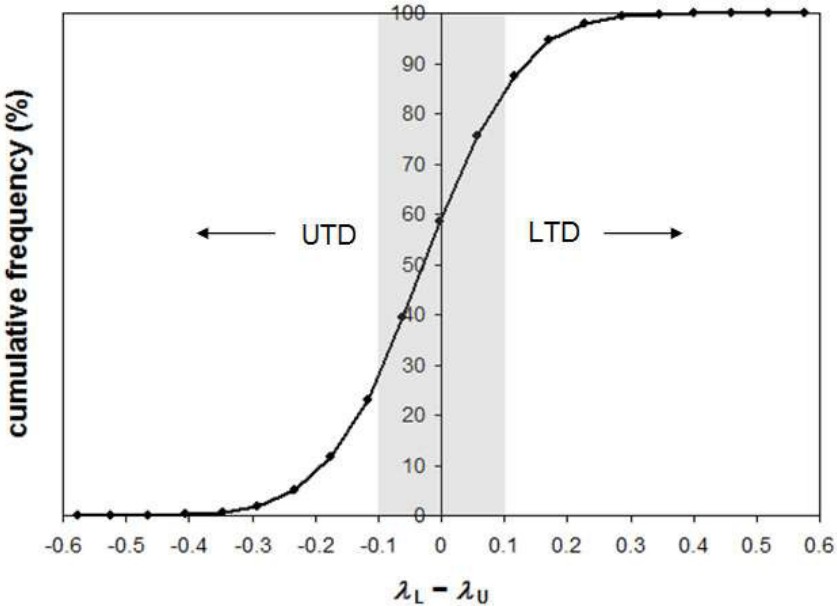


**Fig. 2.** Analysis of the frequency of the empirical tail-dependence coefficients. The plot shows the
cumulative frequency distribution of the differences between the empirical $\lambda_L$ and $\lambda_U$ values
computed according to Schmidt and Stadtmueller (2006). The domain with a roughly symmetric
behavior ($|\lambda_L - \lambda_U| < 0.1$) is highlighted by the grey box area.

The results reported in Fig. 2 were used to divide the entire domain in three categories
(symmetric, LTD and UTD) as depicted in Fig. 3. This map shows evidence of some coherent
spatial patterns, such as the predominance of LTD in southern France, southern Italy, northern
Germany and Denmark, and western Ukraine (among others), and a clustering of UTD in Poland,
Czechia, southern Scandinavia, and Greece. The symmetric condition seems overall more spread
across the entire domain, also thanks to the higher frequency, with a slightly predominance over
northern Europe (i.e., northern Scandinavian peninsula and Iceland).

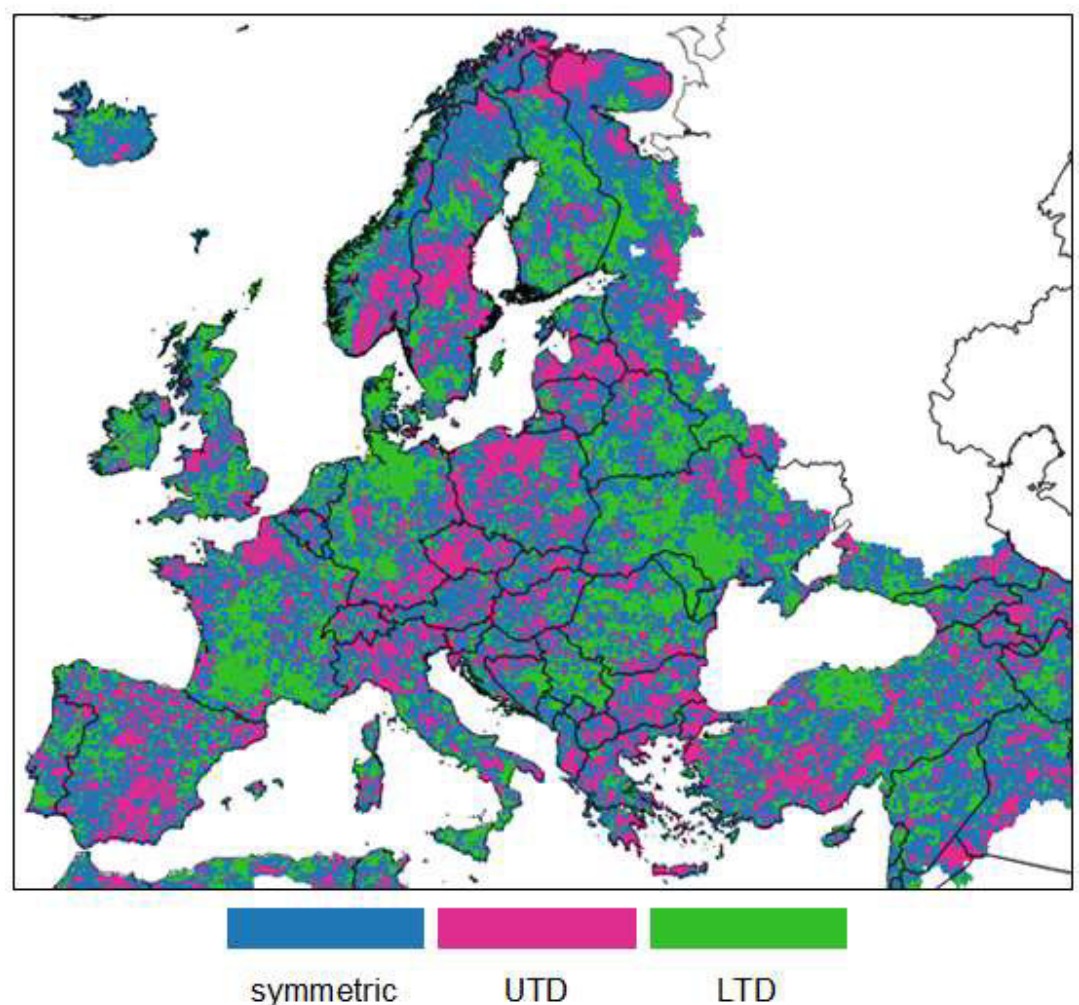

**Fig. 3.** Spatial distribution of the three categories derived from the differences in the empirical tail-
dependence coefficients.

Given the results of the tail-dependence assessment, it is useful to focus the copula
parametric analysis on the capability to reproduce such patterns instead of finding the single copula
that can perform reasonably well over the entire domain. Indeed, the search for the optimal copula
based on the minimum AIC returns the BB7 as the optimal one in about 80% of the domain (not
shown). This result is a consequence of the BB7 flexibility (being derived from a combination of
two purely asymmetric functions), which allows reproducing both symmetric and asymmetric tail-
dependence coefficients according to the values assumed by the two parameters. However, the fact
that a single flexible copula works well over a large range of conditions may hide the key spatial
patterns observed in the TD analysis. These patterns may be better reproduced by adopting a
limited number of more specialized copulas.

By limiting the search to a subset of copula functions, comprising only purely symmetric

or purely asymmetric tail behaviors, more interesting results are obtained, as summarized by the
frequency plot in Fig. 4. The grid cells where symmetric tail behavior copulas are selected as
optimal are about 55% of the domain (see Fig. 4b), with a predominance of Student-t copula but
also with a non-negligible fraction of cells (23%) where the Gaussian (symmetric and without tail-
dependence) is chosen (see Fig. 4a). The remaining grid cells are almost equally split between
upper and lower tail-dependence, with Gumbel (and its rotated counterpart, Gumbel 180) as the
most selected among the asymmetric options.

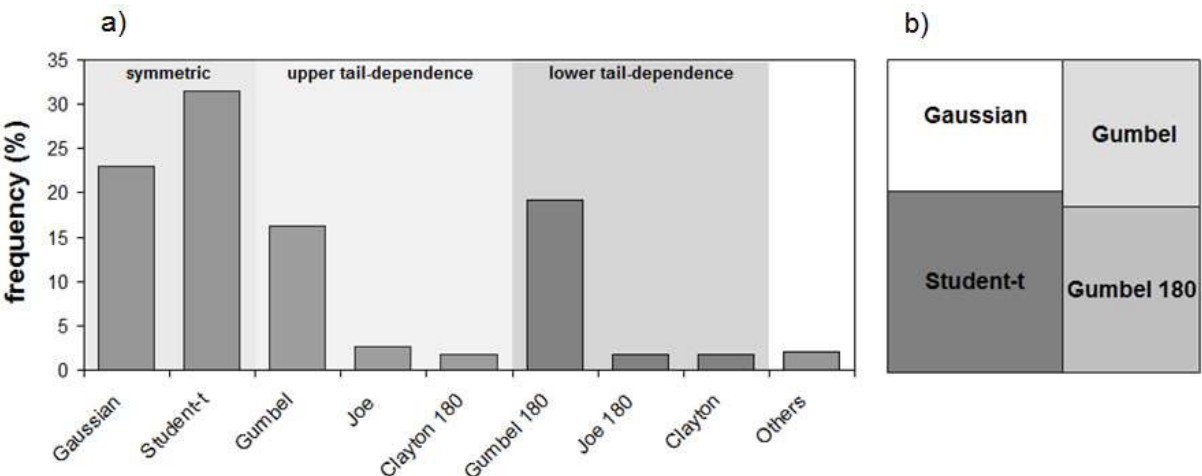


**Fig. 4.** Frequency of the optimal copulas based on the minimum AIC. The barplot in panel a)
shows the frequency of each copula, while the box in panel b) reports a compact description of the
subdivision of the entire domain among the 4 most frequent copulas.

The spatial distribution of these optimal copulas (Fig. 5) mostly agree with the patterns

observed in Fig. 3, supporting the findings on the spatial distribution of TD coefficients. In
addition, this result further confirms that a rather limited range of simple copula functions is able
to capture the overall dynamics of dependence between precipitation and soil moisture over the
entire European domain. Despite the observed spatial clusters in the obtained optimal copulas, the
overall patterns observed in Fig. 5 are still rather noisy and may be difficult to interpret. This erratic
behavior can be partially explained by the fact that different copulas may perform quite similarly
over some grid cells, hence the AIC of the optimal copula ($AIC_{min}$) may not differ significantly
from the AIC of other functions.

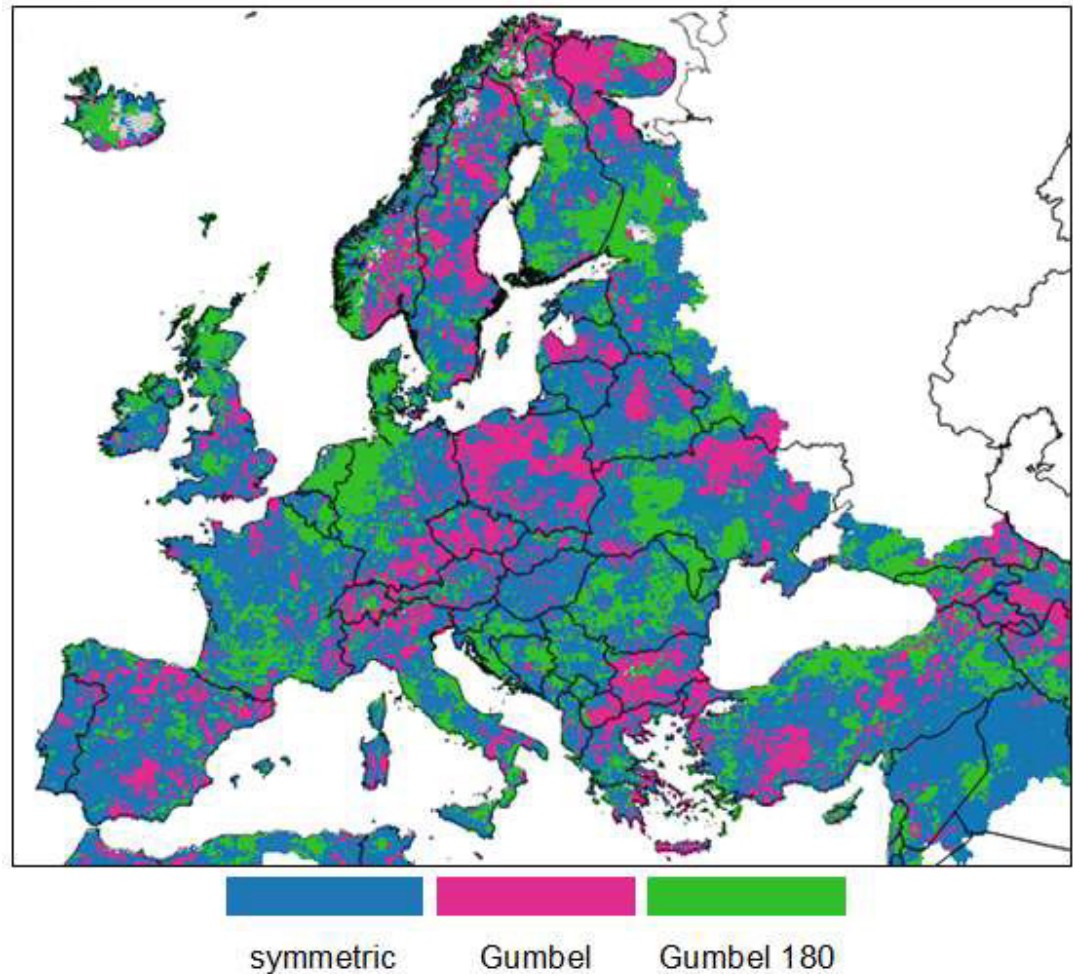


**Fig. 5.** Spatial distribution of the optimal copulas obtained by minimizing the AIC. The symmetric
tail behavior class includes both Gaussian and Student-t copulas.


To further investigate this hypothesis, we evaluated the possibility to replace the optimal
copulas with either a Student-t or a Gumbel (direct and rotated) over the entire domain. The
Gaussian copula was excluded from this analysis under the assumption that the no tail-dependence
of the Gaussian can be adequately reproduced by the Student-t with a small enough tail-
dependence. The plots in Fig. 6 reports the relative likelihood for the Student-t (panel a) and
Gumbel families (panel b) compared to the locally selected optimal copulas. Low values of this
metric correspond to conditions where the optimal copula cannot be replaced by the alternative
function (being either the Student-t or the Gumbel).

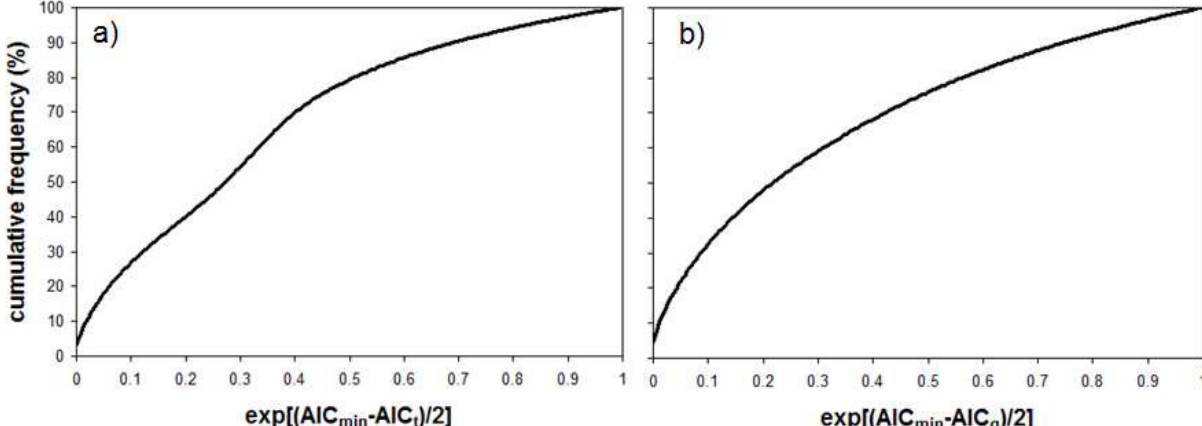


**Fig. 6.** Frequency analysis of the relative likelihood computed between the optimal AIC ($AIC_{min}$)
and: a) Student-t ($AIC_t$), or b) Gumbel ($AIC_g$) families. The grid cells where either the Student-t
or the Gumbel was already the optimal solution were excluded from the respective frequency
analysis.

The results in Fig. 6 show that, if we assume a relative likelihood of 0.1 as a threshold to
detect a statistically significant difference, the Student-t cannot reasonably replace the local
optimal copula in about 18% of the entire domain (Fig. 6a), whereas this fraction is about 17% for
the Gumbel family (Fig. 6b). It emerges that the Gumbel family is the optimal one in almost the
totality (about 99%) of the grid cells where the Student-t is not a suitable replacement of the local
optimal, whereas almost only symmetric copulas (63% Student-t and 34% Gaussian) are the
optimal functions where the Gumbel family is not a suitable replacement. Overall, these results
suggest that the selection of the optimal copula is "univocal" (i.e., cannot be reasonably replaced
by another function) in about 35% (18+17) of the domain, whereas either the Student-t or the
Gumbel families can be adopted in the remain fraction of the domain with similar performances
in terms of AIC (and no clear TD behavior). This analysis also confirms the assumption that all
the areas where the Gaussian was chosen as optimal copula can be satisfactory modeled by using
the Student-t (i.e. without a statistically significant increase in AIC).

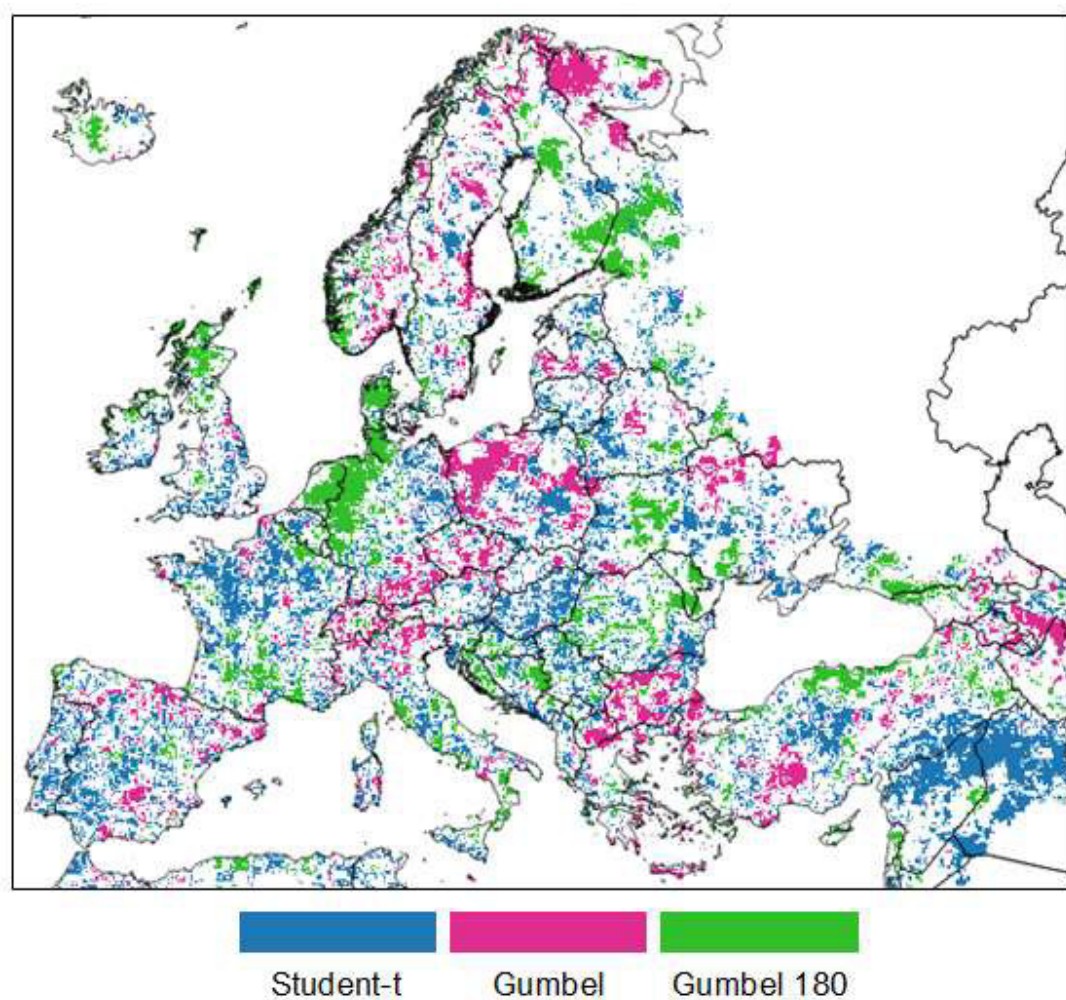

Student-t    Gumbel    Gumbel 180

**Fig. 7.** Spatial distribution of the grid cells where the selection of the optimal copula is "univocal" according to the relative likelihood criterion.

The "univocal" areas derived from the previous analysis are mapped in Fig. 7, highlighting some of the more consistent spatial clusters already observed in both Figs. 3 and 5, as well as a large fraction of cells in northern Europe where a "univocal" optimal copula cannot be selected. These grid cells with "univocal" copula are used as a starting point for the random forest classification, given the robustness in their signal, and the agreement in the outcome of both parametric and non-parametric TD behaviors.

A sample corresponding to 25% of the "univocal" grid cells (about 8% of the entire domain) was used to train the random forest, adopting a number of trees (ntree) of 80 and a single feature randomly sampled at each split (mtry = 1). The training size and the minimum values of

hyperparameters were chosen to reduce the problem of overfitting. Among the possible features,
three variables were selected by analyzing the variable importance plots, as well as the ease of
access: annual average temperature, annual total precipitation, and precipitation seasonality. The
trained classifier was then applied to the testing subset (the remaining 75% of the "univocal" grid
cells) and the outcomes were analyzed by mean of a confusion matrix, which results are
summarized in Table 2. Overall, the obtained classification has a very satisfactory matching with
the test subset, with a general high accuracy (ACC = 0.86) and with all the metrics pointing toward
a significant improving in the performance compared to the reference No-Information-Rate (NIR)
(i.e., small p-values) and a high probability to have correct modeled values compared to simple
chance (i.e., high Cohen's $K$).

**Table 2.** Summary of the confusion matrix analysis applied to the trained random forest on the
testing subset.

| Accuracy (ACC) | 0.86 |
|---|---|
| No-Information-Rate (NIR) | 0.50 |
| p-value (ACC > NIR) | $< 2.2 \times 10^{-16}$ |
| McNemar's test p-value | $3.44 \times 10^{-5}$ |
| Cohen's kappa statistic ($K$) | 0.78 |


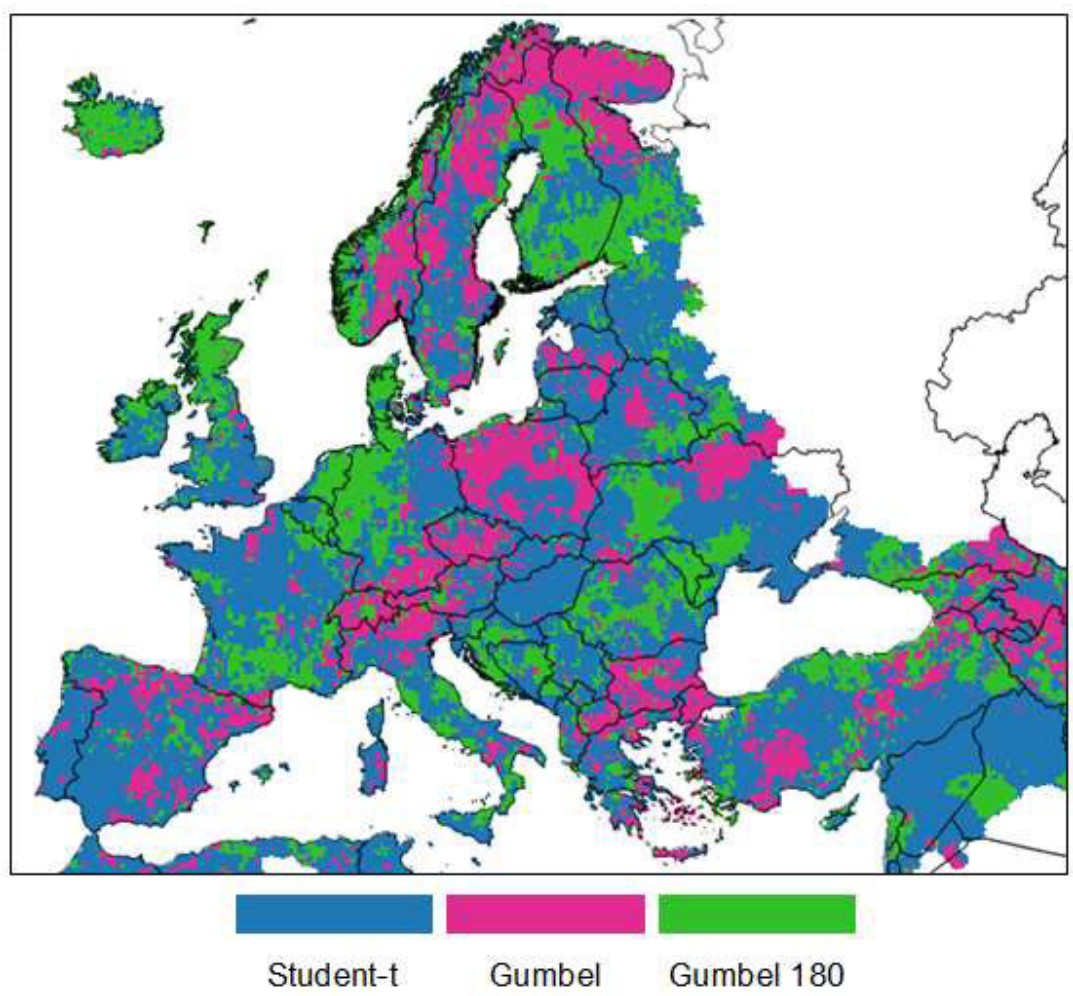

Student-t     Gumbel     Gumbel 180


**Fig. 8.** Map of the optimal copula as modeled by the trained random forest classifier.

Finally, the trained classifier was applied to the entire dataset to obtain a classification of

the European domain in term of the expected optimal copula and the corresponding TD behavior.
This map, reported in Fig. 8, shows a strong resemblance to both the empirically-derived map in
Fig. 3 and the optimal AIC fitting in Fig. 5. Beside this overall agreement, some notable
discrepancies can be observed over northern Scandinavia and Iceland, two regions where low
Kendall's τ and a small fraction of "univocal" selected copulas were already identified.

**4. Discussion**
The overarching goal of the study is to investigate the joint probability of two standardized
variables aiming at capturing agricultural drought conditions, hence the overall agreement between
these two quantities is a fundamental prerequisite. A direct relationship between standardized 3-
month cumulated precipitation and soil moisture is expected, since both SPI-3 and SMA are
similarly-used agricultural drought indices, and this can support the identification of the most
suitable set of copula families (Salvadori et al., 2007; Genest et al., 2007). This direct relationship
is overall confirmed by the positive Kendall's $\tau$ values estimated over most of the domain ($\tau$ =
0.42±0.1). Moderately high correlation values of standardized precipitation and soil moisture were
estimated also in other studies. Kwon et al. (2018) reported Pearson's *r* values between 0.4 and
0.6 for 55 stations in South Korea, albeit with seasonal patterns; Gaona et al. (2022) found similar
values over the Ebro basin with both land-surface modeled and satellite soil moisture, and
Sepulcre-Cantó et al. (2012) obtained an average value of *r* of about 0.6 over nine stations across
Europe.
Sehler et al. (2019) studied the correlation between remote sensing-based precipitation and
soil moisture, finding moderate correlation over southern Europe, and a weak (often not
significant) correlation in central Europe. However, central Europe is close to the upper limit of
the analyzed remote sensing products, which can explain such low performance. Limited
correlation even among different soil moisture products has been observed in northern Europe in
other studies (Almenda-Martín et al., 2022), confirming the difficulty to model soil moisture
dynamics over this region.
The obtained values for the Kendall's $\tau$ fall in a somewhat optimal range for the analysis
of the joint probability, since they are statistically significant almost everywhere (i.e., the two
indices are to a certain degree consistent) but not too high to make meaningless any joint use of
the two datasets (i.e., the two indices are too similar and provide the same information).
The outcome of the tail-dependence analysis is even more interesting, given the role that
such metric plays in the detection of extreme events (and in particular the low-tail for droughts).
The TD investigation is sometimes overlooked in the development of multivariate drought indices,
where previous studies often focused on optimizing the copula to the local data without analyzing
the implicit assumption on the TD, the consistency with the non-parametric TD, and the
implications of the associated dependence. Previous studies on the joint probability of precipitation
and soil moisture are rather scarce, and TD is rarely the focus of such analyses or, at least, limited
to specific areas and/or conditions.
As an example, Manning et al. (2018) performed a very detailed analysis over 11 FluxNet
sites in Europe on the role of precipitation and evapotranspiration on soil moisture drought, based
on pair copula constructions, but the authors did not provide any indication on which bivariate
copula was the optimal one for each site. Kwon et al. (2018) reported that Frank copula was the
most frequent optimal choice in their study over South Korea. However, some clear spatial patterns
observed in their outcomes were not discussed, with Frank being the selected copula mostly in the
central area of the domain, but with Gumbel and Student-t performing the best in the southern and
eastern coasts, respectively.
Dash et al. (2019) found Frank (among the Archimedean copulas) working the best for 3-
month precipitation and soil moisture over an Indian basin; while Hao and AghaKouchak (2013)
highlighted the good performance of Frank and Gumbel in five regions of California, even if
neither Gaussian nor Student-t were considered. In all these applications, no specific
considerations on the TD behaviors were reported, even if a common trend seems to be the good
performance of Frank copula. This is in contrast with our results, where the Frank was very rarely
selected as optimal (less than 1% of the domain). A possible explanation of these results may be
our focus on empirical marginal frequencies rather than theoretical ones, given the well-
documented increasing uncertainty in parametric fitting in the tails (Farahmand and
AghaKouchak, 2015; Laimighofer and Laaha, 2022). As a possible confirmation of this
hypothesis, a good performance of Gumbel and Gaussian has been observed over Iran by Bateni
et al. (2018), similarly to our results, when a non-parametric form for SPI and SSI (Standardized
Soil Moisture Index) was used.
The absence of a strict standard procedure to investigate tail-dependence may be another
factor affecting the limited focus on the topic in many studies on multivariate drought indices.
Non-parametric TD has the clear advantage to avoid any alteration of the data due to the fitting
procedure, but the outcomes in this study also show a high degree of spatial noise likely due to the
intrinsic nature of non-parametric analyses, the large uncertainty in non-parametric methods
(Serinaldi et al., 2015), as well as the effects of the limited sample size (for this last issue see also
the illustration 3.18 in Salvadori et al., 2007). The threshold used here to define a symmetric
behavior, based on a random shuffling of the data, seems to successfully overcome the difficulty
to define a self-consistent maximum difference in TD values, but it cannot be seen as a reliable
approach to easily identify TD symmetry without the support of further evidence (e.g., by
theoretical analyses).

440   In this regard, the fitting of parametric copula functions returns spatial patterns in TD
441 coefficients similar to the ones obtained with the non-parametric approach. However, the absence
442 of "univocal" fittings can be observed for large areas, as well as some contrasting results compared
443 to the non-parametric TD especially over northern Europe (areas with low correlation). The grid
444 cells where a given copula clearly outperforms the alternative options is limited to roughly 1/3 of
445 the domain, further stressing the evidence that clear-cut outcomes are difficult to infer from a
446 single methodology. Thus, it seems reasonable to state that only a critical concerted analysis of
447 both parametric and non-parametric TDs can return robust practical indications based on a
448 converge of evidence.

449   A clear outcome of our study is the predominance of regions with symmetric tail-
450 dependence coefficients, where the Student-t copula is suitable to reproduce the joint probability
451 of standardized precipitation and soil moisture. An even split of the remaining domain between
452 areas with either lower or upper tail-dependence is also observed, where the Gumbel copula (in
453 either is direct or 180 rotated forms) is proven to be a suitable option. These results are crucial in
454 defining the role of standardized precipitation and soil moisture datasets in detecting drought
455 events, and to which extent they can work in synergy in a drought monitoring system. While the
456 correlation between the two datasets highlights the extent of their overall agreement, which in this
457 study was somewhat uniform across most of the domain ($\tau$ ranging between 0.3 and 0.5), very
458 different degrees of tail-consistency can be obtained for similar Kendall's $\tau$ if the TDs differ
459 substantially. Regions with higher LTD will have a higher agreement in the detection of drought
460 extremes compared to areas with a UTD predominance, hence a low number of false alarm and a
461 higher signal-to-noise ratio may be expected.

462   To further explore this behavior, the time series of standardized variables were converted
463 in binary vectors based on the commonly used standardized drought threshold of -1 (corresponding
464 to an empirical frequency of 0.16). On these data, the pair-wise binary correlation coefficient, $\rho$(-
465 1), was computed separately for the grid cells with LTD and UTD. Results are shown in Fig. 9,
466 for grid-cells with low ($0.1 < \tau \leq 0.4$, panel a) and high ($\tau > 0.4$, panel b) overall correlation,
467 respectively. They show a net increase in the pairwise binary correlation for the grid cells with
468 LTD (of about 0.15 in both cases) compared to the cases with UTD, even if the overall correlation
469 is comparable. This increase in $\rho$(-1) translates in a stronger agreement in the detection of extremes
470 when a low tail-dependence is observed, resulting in a more robust detection of the drought

conditions thanks to the concurrency of extreme conditions in both drought indices (i.e.,
convergence of evidence).

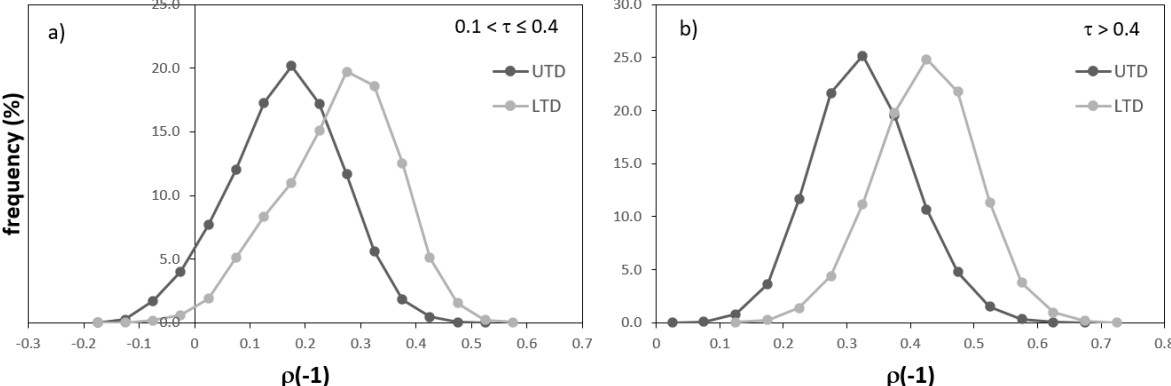

**Fig. 9.** Frequency distribution of the pairwise binary correlation between standardized
precipitation and soil moisture lower than -1, computed separately for grid cells with UTD (dark
grey lines) and LTD (light grey lines). Panel a) reports the results for the grid cells with low overall
correlation ($0.1 < \tau \leq 0.4$), while panel b) reports the results for the grid cells with high correlation
($\tau > 0.4$).

Regions such as southern France, northern UK, northern Germany and Denmark (where a
strong LTD is observed, see Fig. 8) are appropriate candidates for a robust assessment of
agricultural drought conditions based on a joint precipitation-soil moisture index, whereas some
regions in central Europe (i.e., Poland, Czechia, Switzerland) may not equally benefit from the use
of a joint index due to the lower importance of LTD.
Overall, the parametric copula fittings confirm most of the non-parametric TD patterns
suggesting that a parametric approach is suitable for an operational implementation of a
precipitation-soil moisture joint drought index over most of Europe. This implies that the proposed
procedure, based on the combination of parametric and non-parametric analyses, can be considered
a reliable tool to provide meaningful insight on the potential application of joint probability as
detector of extreme droughts.
At first glance, it may seem difficult to assign an explanation for the observed spatial
patterns in LTD and UTD. However, the proven possibility to reasonably reconstruct these spatial
patterns with a random forest classifier, starting from only a small sample of robust training data
(less than 10% of the domain) and with commonly available driving features, suggests that the
observed clusters are unlikely to be caused only by chance and that hidden structures may be
present and may be further explored. This result is encouraging for an extension of the derived
approach to other regions of the world.
**5. Summary and Conclusions**
The use of combined indices based on copula seems a promising development in the field of
drought detection and monitoring. In this study, we analyzed the joint probability of two variables
commonly used in agricultural drought analyses: the empirical frequencies of 3-month cumulated
precipitation and soil moisture. We focus on the probabilistic characteristics being key for
agricultural drought studies.
The overall agreement in the marginal probability of the two standardized variables
suggests that they are indeed valid candidates for the development of a joint drought index over
the European domain. However, an in-depth analysis of the tail-dependence, derived with both
non-parametric and parametric approaches, shows some clear spatial patterns, which have direct
repercussion on the capability of such data to provide robust and coherent estimates of drought
extremes. In this regard, regions such as southern France, northern UK, northern Germany, and
Denmark may benefit more from the joint use of the two standardized variables thanks to the
observed strong low tail-dependence (i.e., increasing agreement on the left tail extremes). The joint
dependence of standardized precipitation and soil moisture is well reproduced by using three
common copulas (Student-t, Gumbel and 180 rotated Gumbel), with spatial patterns that were
successfully reconstructed with a random forest classification, suggesting the presence of a
structure in the outcomes not related to chance.
**Code availability:** The codes used for this analysis can be provided upon request via the
corresponding author.
**Data availability:** All the data used in this study can be accessed and retrieved through the
European Drought Observatory (EDO) web portal
(https://edo.jrc.ec.europa.eu/gdo/php/index.php?id=2112).

**Author contribution:** CC designed the experiments, with inputs from AT and CDM. CC
developed the codes and performed the analyses. CC prepared the manuscript, which was
expanded and revised by all co-authors.

**Competing interests:** At least one of the (co-)authors is a member of the editorial board of
Hydrology and Earth System Sciences.

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
