# Peer review of "Exploring the joint probability of precipitation and soil moisture"

_EGUsphere, 2023_

## Community Comment (CC1)

**Remarks on the manuscript**
**egusphere-2023-1318**

June 29, 2023

I have a few remarks that may help improve the manuscript.

Concerning the limited use of upper/lower tail dependence coefficients in the hydro-climatic literature, I suggest improving the literature review. This would reveal, for instance, that most of the estimators of tail dependence coefficients $\lambda_{\mathrm{L/U}}$ are strongly biased, yielding positive values even if the dependence structure has $\lambda_{\mathrm{L/U}}$ equal to zero. This depends on the fact that these estimators (including Schmidt-Stadtmüller and Capéraà-Fougères-Genest) rely on the implicit or explicit (but not negligible) assumption that the underlying dependence structure is actually characterized by upper tail dependence. In other words, while these estimators may be considered non-parametric in the sense that they do not require the specification of a given copula family, they are strongly parametric in the sense that they require that the underlying copula belongs to a very specific class of models (i.e. those with true tail dependence, basically EV copulas, copulas belonging to EV attraction domain, or similar). These issues are discussed in depth by Serinaldi et al. (2015).

Shuffling procedure should be better explained. If the time series of 3-month precipitation $P$ and $SM$ are shuffled by keeping the correspondence of the observed pairs $(P_i, SM_i)$, this destroys the (possible) serial correlation but keeps the the overall cross-dependence, and therefore summary statistics such as Kendall $\tau_{\mathrm{K}}$ and $\lambda_{\mathrm{L/U}}$. On the other hand, if the shuffling procedure does not retain the pair-wise correspondence between the observed pairs $(P_i, SM_i)$, it destroys the whole cross-dependence structure, not only the upper tail dependence. However, samples resulting from the latter procedure are not informative for the problem at hand. In fact, to build the confidence

intervals (CIs) in Fig. 2, we need samples reproducing all the properties of the observed samples but the tail dependence. Roughly speaking, we need samples keeping e.g. the values of Kendall $\tau_{\mathrm{K}}$ but with $\lambda_{\mathrm{L/U}} = 0$. This is fundamental for a fair assessment of the actual width of the CIs because the above-mentioned estimators of $\lambda_{\mathrm{L/U}}$ are biased, and the estimates of $\lambda_{\mathrm{L/U}}$ are strongly related to the global dependence measured by e.g. Kendall $\tau_{\mathrm{K}}$ (see Serinaldi et al., 2015). If the shuffling procedure keeps the overall cross-dependence removing the upper tail dependence only, therefore the CIs are OK, but the Authors should explain in more detail how they shuffled the data to obtain this effect. Conversely, if the shuffling procedure is just a naïve resampling (bootstrap) destroying any form of dependence, CIs refer to a case which is not comparable with the estimates coming from the observed samples. In other words, CI width is strongly underestimated, and tail symmetry (under sampling uncertainty) cannot be excluded for much more than just the $\cong 50\%$ of locations.

Since the copula-based analysis and modeling reported in the manuscript require independent samples of the pairs $(P_i, SM_i)$ (leaving the above-mentioned bias issues aside), I take for granted that the data are pre-processed to account for seasonality and serial correlation as well as spatial correlation across the region. In this respect, more details about how this is done can help reproduce analysis and results.

Sincerely

Francesco Serinaldi

**References**

Serinaldi, F., Bárdossy, A. Kilsby, C.G. Upper tail dependence in rainfall extremes: would we know it if we saw it?. Stoch Environ Res Risk Assess 29, 12111233 (2015). https://doi.org/10.1007/s00477-014-0946-8

---

## Author Response (AR1)

**Referee #1**

In this work, the authors investigated the joint probability of soil moisture and precipitation over Europe in order to derive meaningful insights on the combined usage of these variables for the detection of agricultural droughts within a probabilistic modelling framework. The in-depth analysis of the tail-dependence especially reveals clear spatial patterns, such as identification of regions which may benefit more from the joint use of the two variables due to the observed strong low tail-dependence, over others. The authors also showed that the spatial patterns are significant using a random forest classification. The scientific goal and the findings of the manuscript are of high relevance and are presented in a clear, concise and well structured way. There are only but a few grammatical errors and typos which could be corrected upon a thorough reading of the manuscript.

We would like to thank the reviewer for the positive feedback on our manuscript. We carefully revised the text to remove errors and typos.

**Referee #2**

General Overview:

The manuscript deals with the investigation of the joint probability of precipitation and soil moisture by using different copula functions and a large dataset over Europe. The analysis of the tail-dependence shows clear spatial patterns in non-parametric and parametric approaches. The manuscript is an interesting approach that could be valuable to drought studies and, presented the approach in a clear and well-structured way. However, I have a few concerns which should be resolved before recommending the paper for publication.

Major remarks:

1) The independence is questionable between the 3-month accumulated precipitation and soil moisture which is a requirement in copula-based analysis, but it can be checked using some statistical tests.

We tested the temporal dependence of each standardized time series by analysing the partial auto corelation function (PACF). The PACF suggests the presence of statistically significant auto correlation only for lag = 1, as somewhat expected for both 3-month accumulated values and soil moisture data. Overall, we did not consider any additional correction necessary as the sample size is good enough, but we added a paragraph describing the results in the revised version of the manuscript.

2) Is this study looking at the joint probability of precipitation and soil moisture or SPI-3 and SMA? This is not clear to me, and I could not see consistency in the manuscript.

We agree that the variables analysed in this study were not clearly introduced in the manuscript. We analysed SPI-3 and SMA, calculated non-parametrically, using empirical frequencies of 3-month precipitation and soil moisture. This was done to avoid any artifact that may be introduced by performing theoretical fittings ahead of the copula analysis.

We better clarified this procedure in the revised version of the manuscript, and we introduced the terms "standardized" precipitation and soil moisture, which is now used consistently through the text.

3) I would suggest authors to add some explanations with the justification for the practical use of the results in agriculture drought studies and drought characterization.

We revised the discussion section and included an example (Fig. 9) on how the tail-dependence may affect drought detection and characterization in practical cases.

**Comments from the community**

We revised the text to clarify how we referred to drought indicators when discussing the lack of focus of the scientific literature on tail dependence. We also added some additional references on the analysis of tail dependence in multivariate studies in hydrology.

We added the outcome of the pairwise binary correlation to further stress on the differences between the areas with upper- and lower- tail dependence, as detected by the combination of parametric and non-parametric methods. This result also has the goal to highlight operational effects of the tail-dependence on the drought detection.